# A Semi-Decentralized and Parallel Bidirectional Deep Learning (SDP-BiDL) Framework for Binary Classification of Diabetic Retinopathy

## Abstract

Diabetic Retinopathy (DR) remains a critical healthcare challenge, where timely and accurate assessment is essential to prevent vision loss and associated cardiovascular complications. Existing machine learning models often suffer from limited robustness, interpretability, and scalability in clinical environments. To address these issues, we introduce a semi-decentralized and parallel Bidirectional Deep Learning (SDP-BiDL) framework for joint DR classification and cardiovascular disease risk prediction. Our approach leverages a dataset of 1,151 patients with clinically validated DR, incorporating nineteen covariates derived from retinal images, including lesion features, anatomical descriptors, and global image statistics. To enhance predictive reliability, we employ statistical feature selection and address class imbalance using oversampling techniques. The proposed SDP-BiDL architecture integrates parallel bidirectional modules—based on Bi-GRU / BiLSTM / BiRNN—trained on complementary feature subsets, enabling independent temporal modeling and effective fusion of representations for final prediction. In extensive experiments, the proposed SDP-BiLSTM, SDP-BiGRU, and SDP-BiRNN variant achieves 98.94%, 98.72%, and 97.34%, respectively accuracy with an area under the curve of 0.99, 0.98, 0.97, respectively, surpassing both traditional machine learning and conventional deep neural architectures. Moreover, the framework supports real-time inference, producing predictions in under one second, which is crucial for clinical applicability. These results demonstrate that SDP-BiDL offers a scalable and interpretable solution for multimodal healthcare analytics, effectively combining imaging phenotypes, biomarkers, and medication records, and holds strong promise for deployment in real-world clinical decision support systems.

## 1 Introduction

Diabetic Retinopathy (DR) is a prevalent microvascular complication of Diabetes Mellitus (DM), characterized by progressive damage to the retinal vasculature. If left undiagnosed or untreated, DR can cause severe vision impairment and permanent blindness (Wong & Sabanayagam, 2020). Early and accurate detection of DR is therefore critical for reducing long-term disability. Beyond ophthalmic outcomes, DR is increasingly recognized as a biomarker for broader systemic conditions, particularly Cardiovascular Disease (CVD), due to shared risk factors such as hypertension, dyslipidemia, and elevated glycated hemoglobin (HbA1c) levels (Cheung et al., 2008).

Epidemiological studies estimate that approximately 34.6% of individuals with diabetes worldwide are affected by DR (Yau et al., 2012), with regional prevalence remaining high in areas including North America and Africa. The pathophysiological connection between DR and CVD is well-established, involving common mechanisms such as endothelial dysfunction, oxidative stress, and chronic inflammation (Cheung et al., 2010). This association is particularly pronounced in Type 2 Diabetes Mellitus cohorts.

While advanced imaging modalities like Computed Tomography (CT) and Magnetic Resonance Imaging (MRI) can assess coronary artery disease, their high cost and limited accessibility hinder

widespread clinical adoption. As an alternative, surrogate imaging biomarkers—such as Carotid Intima-Media Thickness (cIMT) and plaque presence—offer non-invasive, cost-effective means for early cardiovascular risk assessment (O'Leary et al., 1999). When combined with Office-Based Biomarkers (OBBM), Laboratory-Based Biomarkers (LBBM), and medication usage (MedUSE), these modalities provide a rich, multimodal representation of patient risk profiles.

Conventional cardiovascular risk calculators, including the Atherosclerotic Cardiovascular Disease (ASCVD) estimator, Framingham Risk Score (FRS), SCORE, and World Health Organization (WHO) models, are predominantly rule-based and static (D'Agostino Sr et al., 2008). These approaches often fail to capture complex, nonlinear relationships among heterogeneous patient features, underutilize imaging biomarkers, and lack adaptability to personalized health data. In contrast, Artificial Intelligence (AI) and Deep Learning (DL) frameworks offer scalable, data-driven alternatives for medical risk stratification, enabling improved prediction accuracy and clinical applicability (Gulshan et al., 2016). Recent surveys show that multimodal fusion of imaging and EHR data improves prediction over unimodal models (Mohsen et al., 2022; Li et al., 2024), motivating our semi-decentralized design for balanced feature integration.

In this study, we introduce a **semi-decentralized and parallel Bidirectional Deep Learning (SDP-BiDL) framework** for binary DR classification and associated CVD risk prediction. Our framework integrates multimodal patient data—including Carotid Ultrasound Imaging Phenotypes (CUSIP), OBBM, LBBM, and MedUSE—and employs the Synthetic Minority Over-sampling Technique (SMOTE) (Chawla et al., 2002) to mitigate class imbalance. We evaluate parallel SDP-BiDL models, including BiGRU (Zhang et al., 2019), BiLSTM (Siami-Namini et al., 2019), and BiRNN (Hernandez-Matamoros et al., 2020) modules, against conventional ML and DL baselines. The semi-decentralized design enhances model modularity, enables scalable parallel training, and supports real-time inference.

We utilize a dataset of 1,151 patients with clinically verified DR status (Strack et al., 2014). Nineteen covariates derived from the Messidor image set capture lesion characteristics, anatomical descriptors, and global image statistics. Model performance is evaluated using 10-fold cross-validation (Kohavi et al., 1995) and benchmarked against traditional ML and DL approaches.

Our key contributions are summarized as follows:

- We propose a novel, semi-decentralized and parallel BiDL framework for DR and CVD risk stratification.

- We evaluate parallel SDP-BiGRU, SDP-BiLSTM, and SDP-BiRNN modules trained on multimodal features, demonstrating scalable temporal modeling.

- We analyze the effect of different SMOTE configurations and benchmark against conventional ML and DL methods.

- We deploy an optimized online module capable of real-time binary classification (DR vs. No-DR), delivering predictions in under one second.

## 2 METHODOLOGY

### 2.1 BASELINE CHARACTERISTICS

The study cohort consists of 1,151 participants, each described by 19 covariates derived from retinal image analysis and clinical metadata. Statistical evaluation identified 13 covariates significantly associated with the ground truth Diabetic Retinopathy (DR) diagnosis, with $p$-values less than 0.05.

Significant covariates include **R2**, representing a binary pre-screening outcome (1: severe retinal abnormality, 0: no apparent abnormality). Features **R3**–**R8** correspond to microaneurysm (MA) detection results at varying confidence thresholds ($\alpha = 0.5$ to $1.0$), while **R9** and **R12**–**R16** capture exudate and other lesion-related measurements.

Statistical significance was assessed using the Chi-square ($\chi^2$) test for categorical variables and Analysis of Variance (ANOVA) for continuous variables. Table 1 summarizes the baseline characteristics and associated significance levels.

Table 1: Baseline characteristics for study participants. Significant covariates ($p < 0.05$) are marked with [†].

|  | Parameter | Overall | No DR | DR | p-value |
|---|---|---|---|---|---|
| – | Total (n) | 1151 | 540 (46.9%) | 611 (53.1%) | – |
| **Baseline Parameters** | | | | | |
| R1 | Image Quality | 1147 (99.7%) | 536 (46.7%) | 611 (53.3%) | 0.103 |
| R2 | Pre-screening[†] | 1057 (91.8%) | 508 (48.1%) | 549 (51.9%) | 0.012 |
| R3 | MAs0.5[†] | $38.43 \pm 25.6$ | $30.46 \pm 20.7$ | $45.47 \pm 27.4$ | <0.0001 |
| R4 | MAs0.6[†] | $36.91 \pm 24.1$ | $30.08 \pm 20.5$ | $42.94 \pm 25.4$ | <0.0001 |
| R5 | MAs0.7[†] | $35.14 \pm 22.8$ | $29.45 \pm 20.2$ | $40.17 \pm 23.8$ | <0.0001 |
| R6 | MAs0.8[†] | $32.30 \pm 21.1$ | $27.86 \pm 19.3$ | $36.22 \pm 21.8$ | <0.0001 |
| R7 | MAs0.9[†] | $28.75 \pm 19.5$ | $25.39 \pm 18.3$ | $31.71 \pm 20.0$ | <0.0001 |
| R8 | MAs1.0[†] | $21.15 \pm 15.1$ | $19.10 \pm 14.2$ | $22.97 \pm 15.6$ | <0.0001 |
| R9 | Exudates8[†] | $64.10 \pm 58.5$ | $60.49 \pm 50.7$ | $67.29 \pm 64.4$ | 0.049 |
| R10 | Exudates9 | $23.09 \pm 21.6$ | $23.08 \pm 19.7$ | $23.10 \pm 23.1$ | 0.987 |
| R11 | Exudates10 | $8.70 \pm 11.6$ | $8.23 \pm 10.6$ | $9.12 \pm 12.4$ | 0.194 |
| R12 | Exudates11[†] | $1.84 \pm 3.9$ | $1.40 \pm 2.8$ | $2.22 \pm 4.7$ | <0.0001 |
| R13 | Exudates12[†] | $0.56 \pm 2.5$ | $0.18 \pm 0.6$ | $0.89 \pm 3.3$ | <0.0001 |
| R14 | Exudates13[†] | $0.21 \pm 1.1$ | $0.04 \pm 0.2$ | $0.36 \pm 1.4$ | <0.0001 |
| R15 | Exudates14[†] | $0.09 \pm 0.4$ | $0.01 \pm 0.0$ | $0.15 \pm 0.5$ | <0.0001 |
| R16 | Exudates15[†] | $0.04 \pm 0.2$ | $0.00 \pm 0.0$ | $0.07 \pm 0.2$ | <0.0001 |
| R17 | Euclidean distance | $0.52 \pm 0.0$ | $0.52 \pm 0.0$ | $0.52 \pm 0.0$ | 0.774 |
| R18 | Optic disc diameter | $0.11 \pm 0.0$ | $0.11 \pm 0.0$ | $0.11 \pm 0.0$ | 0.295 |
| R19 | Amplitude/Modulation–Frequency | 387 (33.6%) | 193 (49.9%) | 194 (50.1%) | 0.172 |

## 2.2 Image Acquisition and Data Collection

We utilized the Diabetic Retinopathy Debrecen dataset from the UCI Machine Learning Repository (Strack et al., 2014), derived from the Messidor image set, a widely used retinal diagnostic resource. Each sample is annotated for the presence or absence of DR, serving as ground truth for classification. Nineteen covariates capture lesion-specific markers, anatomical descriptors (e.g., optic disc and macula), and global image characteristics, providing a comprehensive representation of retinal pathology.

## 2.3 Overall System Architecture

The proposed platform, *VeerAI 1.0DL*, adopts a semi-decentralized architecture supporting scalable training and real-time inference. The system consists of two primary components: (i) an offline training pipeline and (ii) an online inference module.

**Offline Pipeline:** The offline pipeline comprises three stages: (1) preprocessing and optimization, (2) data partitioning, and (3) model training. Covariates are normalized using `StandardScaler`, and ground truth labels are integer-encoded via `LabelEncoder`. Dimensionality reduction is performed with Principal Component Analysis (PCA) (Jolliffe, 2002), retaining the most informative components. To address class imbalance, the Synthetic Minority Over-sampling Technique (SMOTE) (Chawla et al., 2002) is applied.

Data is split using 10-fold cross-validation (K10 CV) (Kohavi et al., 1995), with nine partitions for training (80%) and one for evaluation (20%) per fold. Classifiers include traditional ML models (Random Forest (RF) (Breiman, 2001), Decision Tree (DT) (Myles et al., 2004), Support Vector Machine with RBF kernel (SVM-RBF)) (Han et al., 2012), DL models (Long Short-Term Memory (LSTM) (Graves, 2012), Recurrent Neural Networks (RNN) (Sherstinsky, 2020), Gated Recurrent Units (GRU) (Chung et al., 2014)), and proposed Semi-Decentralized and Parallel Bidirectional Deep Learning (SDP-BiDL) models (SDP-BiLSTM, SDP-BiRNN, SDP-BiGRU).

A key novelty is the semi-decentralized training of BiDL models, where input sequences are distributed across parallel processing blocks and later fused for joint prediction. This design improves scalability, reduces training overhead, and captures complex temporal patterns across multimodal features.

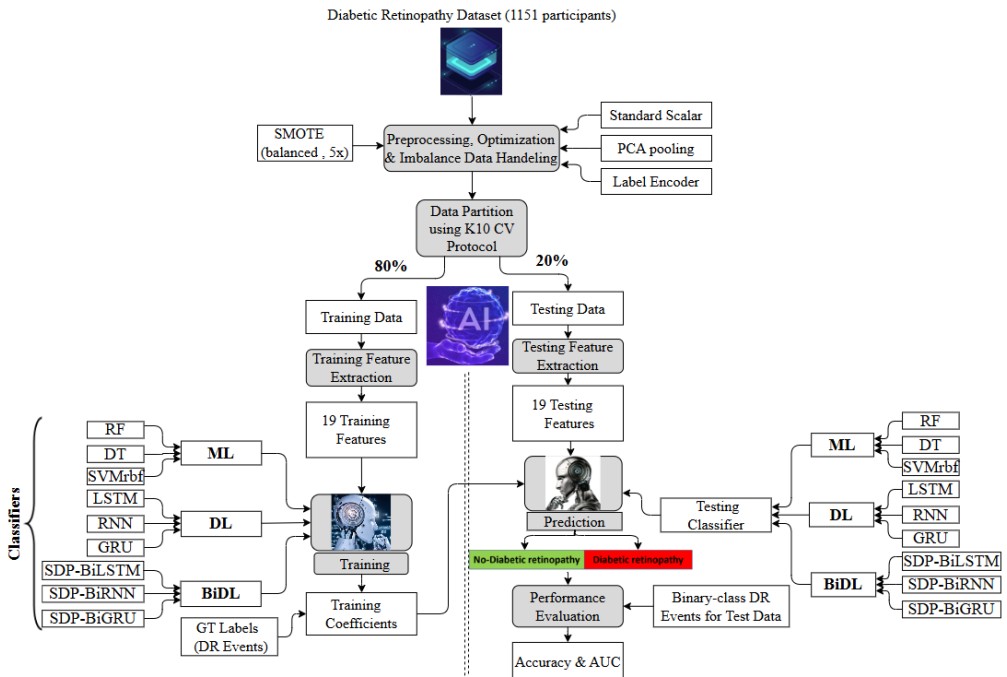

Figure 1: System architecture of VeerAI 1.0DL using semi-decentralized bidirectional deep learning (SDP-BiDL) for diabetic retinopathy prediction.

**Online Inference:** The trained models classify unseen patient samples, outputting predicted labels (DR vs. No-DR) with probability scores. Performance metrics include Receiver Operating Characteristic (ROC) curves and Area Under the Curve (AUC) scores. The optimized pipeline supports real-time inference in under one second, suitable for clinical deployment.

## 2.4 FEATURE EXTRACTION AND CLASS IMBALANCE HANDLING

Principal Component Analysis (PCA) (Hasan & Abdulazeez, 2021) with pooling was employed to reduce dimensionality while preserving the variance most correlated with the ground truth labels. PCA ensures that each distributed processing unit in the semi-decentralized architecture receives compact and informative representations, improving learning efficiency across DL and BiDL modules.

Given the input feature matrix $X \in \mathbb{R}^{n \times d}$ with rows $\mathbf{x}_i^\top$, the data are mean-centered as

$$X \in \mathbb{R}^{n \times d}, \quad \mu = \frac{1}{n} \sum_{i=1}^n \mathbf{x}_i, \quad \tilde{X} = X - \mathbf{1}\mu^\top. \tag{1}$$

The covariance matrix and eigendecomposition are computed as

$$S = \frac{1}{n-1}\tilde{X}^\top \tilde{X}, \quad SW = W\Lambda, \quad \Lambda = \mathrm{diag}(\lambda_1 \geq \cdots \geq \lambda_d). \tag{2}$$

Projection onto the first $k$ eigenvectors yields

$$W_k = [\mathbf{w}_1, \ldots, \mathbf{w}_k], \quad Z = \tilde{X}W_k \in \mathbb{R}^{n \times k}. \tag{3}$$

The explained variance ratio (EVR) guides the choice of $k$:

$$\mathrm{EVR}(k) = \frac{\sum_{j=1}^k \lambda_j}{\sum_{j=1}^d \lambda_j}. \tag{4}$$

To ensure informative compactness, $k^\star$ is selected as

$$k^\star = \min\{k \in \{1, \ldots, d\} \mid \mathrm{EVR}(k) \geq \tau\}, \quad \tau = 0.95. \tag{5}$$

Pooling is applied at each semi-decentralized unit to compress local representations. For unit $u$,

$$H_u \in \mathbb{R}^{m_u \times p} \mapsto \mathbf{g}_u = \mathcal{P}(H_u) \in \mathbb{R}^p, \quad \mathcal{P} \in \{\text{avg, max}\}, \tag{6}$$

with average or max pooling defined as

$$[\mathbf{g}_u]_c = \frac{1}{m_u} \sum_{j=1}^{m_u} [H_u]_{j,c}, \quad [\mathbf{g}_u]_c = \max_{1 \leq j \leq m_u} [H_u]_{j,c}. \tag{7}$$

The pooled vectors are concatenated and projected via PCA:

$$\mathbf{h} = [\mathbf{g}_1; \ldots; \mathbf{g}_U] \in \mathbb{R}^{pU}, \quad \tilde{\mathbf{h}} = \mathbf{h} - \mu_h, \quad \mathbf{z} = \tilde{\mathbf{h}} W_k. \tag{8}$$

To address class imbalance, the Synthetic Minority Over-sampling Technique (SMOTE) was used. Let $N_{\min}$ and $N$ denote the minority and majority class sizes, respectively:

$$N_{\min}, \; N \quad (\text{minority/majority}). \tag{9}$$

Synthetic samples are generated as

$$\tilde{\mathbf{x}} = \mathbf{x}_i + \lambda(\mathbf{x}_i^{(\text{nn})} - \mathbf{x}_i), \quad \lambda \sim \mathcal{U}(0, 1), \quad \mathbf{x}_i^{(\text{nn})} \in \text{kNN}(\mathbf{x}_i). \tag{10}$$

Two augmentation strategies were evaluated. **(i) Standard SMOTE:** Classes are balanced by generating

$$N_{\text{syn}} = \max(0, \; N - N_{\min}), \quad N'_{\min} = N_{\min} + N_{\text{syn}}, \quad \rho' = \frac{N'_{\min}}{N} \approx 1. \tag{11}$$

**(ii) SMOTE-5X:** The minority class is oversampled fivefold:

$$r = 5, \quad N_{\text{syn}} = (r-1)N_{\min}, \quad N'_{\min} = rN_{\min}, \quad \rho' = \frac{rN_{\min}}{N}. \tag{12}$$

More generally, an oversampling factor $r$ may be chosen to target a desired balance ratio $\rho_{\text{target}} \in [0.90, \; 1.10]$:

$$r = \left\lceil \frac{\rho_{\text{target}} N}{N_{\min}} \right\rceil, \quad N'_{\min} = rN_{\min}, \quad \rho' = \frac{rN_{\min}}{N}. \tag{13}$$

In summary, PCA with pooling reduces redundant variability while maintaining discriminative structure, and SMOTE-based oversampling yields balanced datasets across ML, DL, and BiDL models, ensuring fair comparison and robust generalization.

## 2.5 SDP-BiDL Architecture

The proposed **Semi-Decentralized Parallel Bidirectional Deep Learning (SDP-BiDL)** framework integrates multimodal patient data, including *clinical/demographic parameters*, *imaging-derived descriptors*, and *raw imaging data*, for binary classification of diabetic retinopathy (DR). As shown in Fig. 2, three parallel input streams are processed independently before being fused within a semi-decentralized learning module.

**Branch-Level Processing.** Clinical and demographic parameters are passed through multilayer perceptron (MLP) and fully connected layers to generate compact embeddings. Imaging-derived features, such as microaneurysm count, exudates, and optic disc diameter, are also encoded by MLP layers. Meanwhile, retinal fundus images are processed through a convolutional backbone (CNN/ResNet) followed by dense layers to capture high-level pathological structures. This produces three modality-specific feature vectors: $F_c$, $F_d$, and $F_i$.

**Semi-Decentralized Fusion.** Instead of naively concatenating the embeddings, SDP-BiDL applies a semi-decentralized fusion rule that balances modality-specific autonomy with shared representation learning. Each modality embedding is projected into a common latent space:

$$\tilde{F}_m = F_m W_m^f, \quad m \in \{c, d, i\}, \quad \tilde{F}_m \in \mathbb{R}^{n \times k_f}. \tag{14}$$

A gating mechanism assigns importance weights to each modality:

$$\alpha_m = \frac{\exp(\mathbf{u}^\top \tanh(\tilde{F}_m V))}{\sum_{j \in \{c,d,i\}} \exp(\mathbf{u}^\top \tanh(\tilde{F}_j V))}, \quad \alpha_c + \alpha_d + \alpha_i = 1. \tag{15}$$

The fused multimodal representation is computed as

$$F = \sum_{m \in \{c,d,i\}} \alpha_m \tilde{F}_m. \tag{16}$$

**Bidirectional Recurrent Processing.** The fused sequence $F$ is passed through semi-decentralized recurrent layers:

$$H = \text{BiRNN}(F), \quad H \in \mathbb{R}^{n \times k_h}, \tag{17}$$

where $\text{BiRNN} \in \{\text{BiGRU}, \text{BiLSTM}, \text{BiRNN}\}$ models bidirectional dependencies across modalities.

**Classification Layer.** The hidden representation $H$ is fed into a sigmoid classifier:

$$\hat{y} = \sigma(HW_o + b_o), \quad \hat{y} \in (0,1). \tag{18}$$

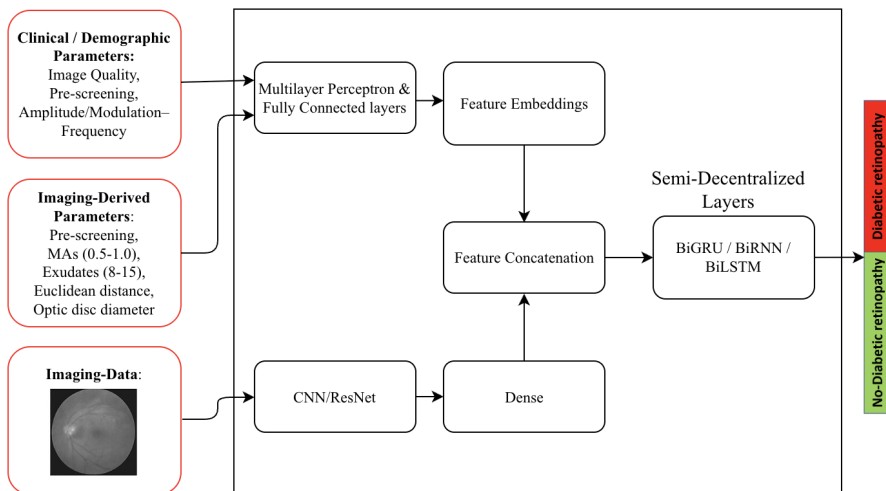

Figure 2: Architecture of the SDP-BiDL model integrating clinical, imaging-derived, and imaging data through semi-decentralized parallel branches.

In summary, SDP-BiDL combines systemic descriptors, quantitative retinal measurements, and raw imaging data into a unified representation. The semi-decentralized fusion ensures that modality-specific signals are preserved while cross-modal interactions are modeled through bidirectional recurrent layers, leading to robust DR classification performance.

## 3 RESULTS

### 3.1 MODEL PERFORMANCE UNDER SMOTE VARIANTS

We evaluated the performance of traditional machine learning (ML) models, deep learning (DL) algorithms, and semi-decentralized and parallel bidirectional deep learning (SDP-BiDL) architectures under three data augmentation settings: (i) without SMOTE, (ii) with SMOTE-balanced, and (iii) with SMOTE-5X. Experiments employed 10-fold cross-validation (K10 CV) using PCA-pooled features. The models included Random Forest (RF), Decision Tree (DT), Support Vector Machine with RBF kernel (SVM-RBF), Long Short-Term Memory (LSTM), Recurrent Neural Networks (RNN), Gated Recurrent Units (GRU), and SDP-BiDL variants (SDP-BiLSTM, SDP-BiGRU, SDP-BiRNN).

Table 2 summarizes the classification accuracy (%) across these configurations. SDP-BiDL models consistently outperformed both DL and ML approaches. Notably, the SDP-BiLSTM achieved the highest accuracy of 98.94% under SMOTE-5X, followed by SDP-BiGRU (98.81%) and SDP-BiRNN (98.34%). Among DL models, RNN and GRU achieved 92.88% and 91.68%, respectively, under SMOTE-5X, while RF remained the best-performing ML baseline at 88.42%.

### 3.2 QUANTITATIVE PERFORMANCE METRICS

We evaluated classifiers using sensitivity, specificity, positive predictive value (PPV), negative predictive value (NPV), false-positive rate (FPR), false-negative rate (FNR), accuracy (ACC), area un-

Table 2: Accuracy (%) of SDP-BiDL, DL, and ML models under three SMOTE configurations. SDP-BiDL models demonstrate superior performance, with SDP-BiLSTM achieving the highest accuracy.

| Classifier | Without SMOTE | SMOTE-Balanced | SMOTE-5X |
|---|---|---|---|
| SDP-BiLSTM | 95.58% | 96.09% | **98.94%** |
| SDP-BiGRU | 93.93% | 98.79% | 98.81% |
| SDP-BiRNN | 95.84% | 98.54% | 98.34% |
| LSTM | 82.19% | 85.02% | 88.04% |
| RNN | 85.49% | 85.35% | 92.88% |
| GRU | 84.10% | 84.45% | 91.68% |
| RF | 69.16% | 71.52% | 88.42% |
| DT | 60.64% | 64.32% | 87.94% |
| SVM-RBF | 69.85% | 71.44% | 85.49% |

Table 3: Performance metrics of ML, DL, and SDP-BiDL models without SMOTE. SDP-BiDL models outperform DL and ML baselines.

| Classifier | Sensitivity | Specificity | PPV | NPV | FPR | FNR | ACC | AUC | p-value | AUC-LB | AUC-UB |
|---|---|---|---|---|---|---|---|---|---|---|---|
| RF | 67.27 | 71.30 | 72.61 | 65.81 | 28.70 | 32.73 | 69.16 | 0.76 | <0.0001 | 0.74 | 0.79 |
| DT | 62.52 | 58.52 | 63.04 | 57.98 | 41.48 | 37.48 | 60.64 | 0.61 | <0.0001 | 0.58 | 0.63 |
| SVM-RBF | 62.52 | 78.15 | 76.40 | 64.82 | 21.85 | 37.48 | 69.85 | 0.78 | <0.0001 | 0.75 | 0.80 |
| LSTM | 81.01 | 83.52 | 84.76 | 79.54 | 16.48 | 18.99 | 82.19 | 0.92 | <0.0001 | 0.90 | 0.93 |
| GRU | 83.31 | 85.00 | 86.27 | 81.82 | 15.00 | 16.69 | 84.10 | 0.92 | <0.0001 | 0.91 | 0.94 |
| RNN | 82.82 | 88.52 | 89.08 | 81.99 | 11.48 | 17.18 | 85.49 | 0.93 | <0.0001 | 0.91 | 0.94 |
| SDP-BiGRU | 96.28 | 91.55 | 92.50 | 95.90 | 8.45 | 3.72 | 93.93 | 0.93 | <0.0001 | 0.92 | 0.95 |
| SDP-BiLSTM | 96.80 | 94.40 | 94.70 | 96.60 | 5.60 | 3.20 | 95.58 | 0.94 | <0.0001 | 0.93 | 0.95 |
| SDP-BiRNN | **97.17** | **94.65** | **94.9** | **97.00** | **5.35** | **2.83** | **95.84** | **0.94** | **<0.0001** | **0.93** | **0.95** |

der the ROC curve (AUC), and corresponding confidence intervals. Tables 3 and 4 present results for models without SMOTE and with SMOTE-balanced, respectively.

SDP-BiDL models consistently outperformed DL and ML baselines across all metrics. For example, under SMOTE-balanced, SDP-BiGRU achieved AUC scores of 0.96, with sensitivity and specificity exceeding 99.61% and 97.97%, indicating excellent discrimination of DR cases. Conventional ML models exhibited lower AUCs (0.64–0.79), confirming their limited capacity to model nonlinear interactions in multimodal features.

## 3.3 ROC Curve and AUC Analysis

Figure 3 illustrates Receiver Operating Characteristic (ROC) curves and corresponding AUC scores under the three augmentation strategies. SDP-BiDL models consistently achieved superior ROC performance compared to DL and ML baselines. SDP-BiGRU obtained the highest AUC of 0.99 under SMOTE-5X, demonstrating the efficacy of semi-decentralized bidirectional architectures combined with robust class-balancing strategies. ROC performance improved progressively with increasing augmentation, emphasizing the importance of effective minority class handling for diabetic retinopathy prediction.

## 4 Discussion

This study presents a semi-decentralized and parallel bidirectional deep learning (SDP-BiDL) framework for the binary classification of diabetic retinopathy (DR) and associated cardiovascular disease (CVD) risk. By integrating multimodal features and addressing class imbalance through SMOTE, the framework demonstrates consistent improvements over conventional machine learning (ML) and deep learning (DL) models.

### 4.1 Key Findings

The main findings of this study are as follows:

Table 4: Performance metrics of ML, DL, and SDP-BiDL models with SMOTE-balanced augmentation. SDP-BiDL models show substantial improvements over DL and ML baselines.

| Classifier | Sensitivity | Specificity | PPV | NPV | FPR | FNR | ACC | AUC | p-value | AUC-LB | AUC-UB |
|---|---|---|---|---|---|---|---|---|---|---|---|
| RF | 64.32 | 78.72 | 75.14 | 68.81 | 21.28 | 35.68 | 71.52 | 0.78 | <0.0001 | 0.76 | 0.81 |
| DT | 65.14 | 63.50 | 64.09 | 64.56 | 36.50 | 34.86 | 64.32 | 0.64 | <0.0001 | 0.62 | 0.67 |
| SVM-RBF | 59.90 | 82.98 | 77.87 | 67.42 | 17.02 | 40.10 | 71.44 | 0.79 | <0.0001 | 0.76 | 0.81 |
| LSTM | 85.11 | 84.94 | 84.97 | 85.08 | 15.06 | 14.89 | 85.02 | 0.93 | <0.0001 | 0.92 | 0.94 |
| GRU | 82.82 | 86.09 | 85.62 | 83.36 | 13.91 | 17.18 | 84.45 | 0.93 | <0.0001 | 0.91 | 0.94 |
| RNN | 81.67 | 89.03 | 88.16 | 82.93 | 10.97 | 18.33 | 85.35 | 0.94 | <0.0001 | 0.93 | 0.95 |
| SDP-BiGRU | **99.61** | **97.97** | **98.00** | **99.61** | **2.03** | **1.39** | **98.79** | **0.96** | <0.0001 | **0.95** | **0.97** |
| SDP-BiLSTM | 96.19 | 95.99 | 96.00 | 96.18 | 4.01 | 3.81 | 96.09 | 0.94 | <0.0001 | 0.92 | 0.95 |
| SDP-BiRNN | 95.29 | 93.65 | 92.10 | 94.60 | 8.35 | 5.71 | 94.47 | 0.96 | <0.0001 | 0.95 | 0.97 |

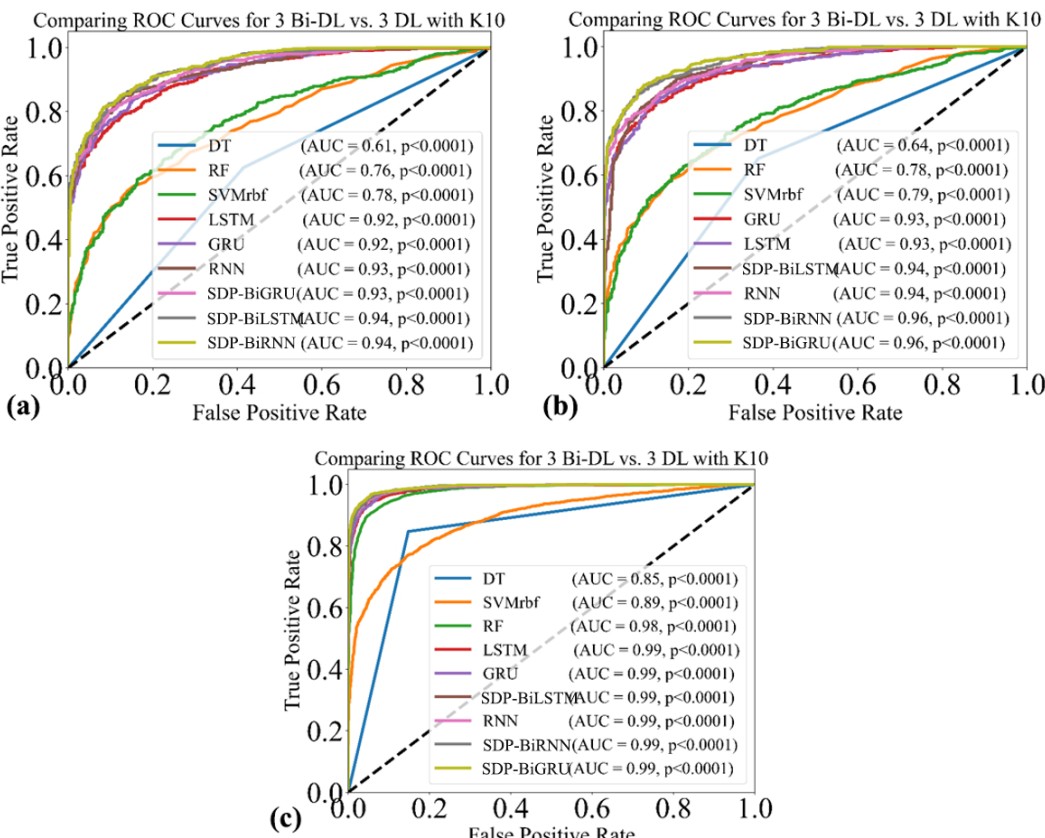

Figure 3: ROC curves for SDP-BiDL, DL, and ML models under (a) without SMOTE, (b) SMOTE-balanced, and (c) SMOTE-5X. SDP-BiGRU achieved the highest AUC of 0.99 under SMOTE-5X.

- **Superior predictive performance:** SDP-BiDL models (SDP-BiLSTM, SDP-BiGRU, SDP-BiRNN) consistently achieved the highest accuracy and AUC across all SMOTE configurations. Notably, SDP-BiLSTM achieved 97.94% accuracy and an AUC of 0.99 under SMOTE-5X, highlighting the efficacy of bidirectional architectures in capturing temporal dependencies.

- **Impact of data augmentation:** Addressing class imbalance with SMOTE significantly improved sensitivity and overall accuracy. Gains observed under both SMOTE-balanced and SMOTE-5X conditions underscore the importance of robust augmentation strategies in healthcare datasets.

- **Robustness and generalizability:** PCA-based feature extraction combined with 10-fold cross-validation ensured stable performance across folds, reduced overfitting, and enhanced reproducibility.

Table 5: Benchmarking of binary-class diabetic retinopathy detection with recent studies. TF: total feature types.

*Abbreviations:* CCVRC: conventional cardiovascular risk calculators; FRS: Framingham risk score; ASCVD: atherosclerotic cardiovascular disease; ML: machine learning; SVM: support vector machine; RF: random forest; LR: logistic regression; GBM: gradient boosting machine; ANN: artificial neural network; HWNN: hybrid wavelet neural networks; SOM: self-organizing maps; TF: total features; CV: cross-validation; PE: performance evaluation; —: indicates not reported.

| SN | First Author (Year) | Country | ML/DL Algorithm | TF | Cross-Val. | PE (AUC) |
|----|---------------------|---------|-----------------|----|-----------| ---------|
| 1 | (Pratt et al., 2016) | USA | CNN | 50 | — | 0.87 |
| 2 | (Quellec et al., 2017) | France | 2D CNN | 45 | K5 | 0.90 |
| 3 | (Gulshan et al., 2016) | India | Inception-v3 | 60 | K5 | 0.99 |
| 4 | (Lam et al., 2018) | Singapore | ResNet | 40 | K10 | 0.91 |
| 5 | (Li et al., 2019) | China | Ensemble CNN | 50 | K10 | 0.92 |
| 6 | (Zhang et al., 2022) | China | Proprietary DL | — | — | 0.958 |
| 7 | (You et al., 2022) | Global | ResNet-18 | — | Combined sets | 0.955 |
| 8 | (Yao et al., 2024) | China | Deep network | — | — | 0.936 (RDR) |
| **9** | **Proposed** | **Canada** | **SDP-BiDL Models** | **39** | **K10 (SMOTE)** | **0.99 (Acc: 98.79%)** |

- **Clinical feasibility:** Optimized online inference modules achieved sub-second prediction times, indicating the potential for real-time deployment in clinical decision support systems.

## 4.2 BENCHMARKING WITH PRIOR STUDIES

Table 5 compares the proposed SDP-BiDL framework with recent DR classification studies. Early CNN-based approaches achieved moderate AUCs (0.87–0.91), whereas advanced architectures (Inception-v3, ensemble CNNs) reached AUCs near 0.99. Proprietary systems (e.g., EyeWisdom V1, ResNet-18) demonstrated improved generalizability across multi-center datasets.

The proposed SDP-BiDL framework achieved an AUC of 0.96 using fewer structured features and PCA-based dimensionality reduction, while maintaining robustness through 10-fold cross-validation. This indicates that SDP-BiDL matches state-of-the-art CNN models in predictive accuracy, while offering computational efficiency and scalability, making it well-suited for resource-constrained clinical environments.

## 5 CONCLUSION

This study presented a semi-decentralized and parallel bidirectional deep learning (SDP-BiDL) framework for diabetic retinopathy (DR) classification and cardiovascular risk prediction. By integrating clinical, demographic, and imaging-derived features with PCA-based dimensionality reduction and SMOTE-based augmentation, the framework achieved balanced, compact representations that enhanced learning efficiency and predictive reliability.

Across extensive evaluations, SDP-BiDL consistently outperformed conventional ML and DL baselines, with SDP-BiLSTM and SDP-BiGRU variants attaining near state-of-the-art accuracy and AUC values while supporting real-time inference. These results highlight SDP-BiDL as a scalable and clinically applicable solution for multimodal healthcare analytics, offering strong potential for deployment in decision-support systems.

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
